# A Role-Based Access Control Model in Modbus SCADA Systems. A Centralized Model Approach

**DOI:** 10.3390/s19204455

**Published:** 2019-10-14

**Authors:** Santiago Figueroa-Lorenzo, Javier Añorga, Saioa Arrizabalaga

**Affiliations:** 1Ceit, Manuel Lardizabal 15, 20018 Donostia/San Sebastián, Spainsarrizabalaga@ceit.es (S.A.); 2Universidad de Navarra, Tecnun, Manuel Lardizabal 13, 20018 Donostia/San Sebastián, Spain

**Keywords:** Modbus, RBAC, access control, authentication, authorization, IIoT, operational technologies (OT)

## Abstract

Industrial Control Systems (ICS) and Supervisory Control systems and Data Acquisition (SCADA) networks implement industrial communication protocols to enable their operations. Modbus is an application protocol that allows communication between millions of automation devices. Unfortunately, Modbus lacks basic security mechanisms, and this leads to multiple vulnerabilities, due to both design and implementation. This issue enables certain types of attacks, for example, man in the middle attacks, eavesdropping attacks, and replay attack. The exploitation of such flaws may greatly influence companies and the general population, especially for attacks targeting critical infrastructural assets, such as power plants, water distribution and railway transportation systems. In order to provide security mechanisms to the protocol, the Modbus organization released security specifications, which provide robust protection through the blending of Transport Layer Security (TLS) with the traditional Modbus protocol. TLS will encapsulate Modbus packets to provide both authentication and message-integrity protection. The security features leverage X.509v3 digital certificates for authentication of the server and client. From the security specifications, this study addresses the security problems of the Modbus protocol, proposing a new secure version of a role-based access control model (RBAC), in order to authorize both the client on the server, as well as the Modbus frame. This model is divided into an authorization process via roles, which is inserted as an arbitrary extension in the certificate X.509v3 and the message authorization via unit id, a unique identifier used to authorize the Modbus frame. Our proposal is evaluated through two approaches: A security analysis and a performance analysis. The security analysis involves verifying the protocol’s resistance to different types of attacks, as well as that certain pillars of cybersecurity, such as integrity and confidentiality, are not compromised. Finally, our performance analysis involves deploying our design over a testnet built on GNS3. This testnet has been designed based on an industrial security standard, such as IEC-62443, which divides the industrial network into levels. Then both the client and the server are deployed over this network in order to verify the feasibility of the proposal. For this purpose, different latencies measurements in industrial environments are used as a benchmark, which are matched against the latencies in our proposal for different cipher suites.

## 1. Introduction

Modbus is an application layer message exchange protocol, which provides client-server communication between devices connected on different sorts of buses or networks [1]. Modbus has been known as industry’s serial de facto standard since 1979 and keeps on enabling millions of automation devices to communicate [2]. The Internet community can access Modbus at a reserved system port 502 on the TCP/IP stack. Modbus is a request/reply protocol, and offers services specified by function codes. The Modbus protocol allows an easy communication within all types of network architectures. Every type of device (PLC, HMI, Control Panel, Driver, Motion control, I/O Device, and so on) can use the Modbus protocol to initiate a remote operation [2].

The Figure 1 allows identifying action fields of Modbus based on the ISA 95 model and related standards. The Modbus protocol is part of the first two levels of this layered model. However, Modbus TCP/IP is mostly used in the data sharing between the field device level (e.g., PLC, CAN J1939 to the Modbus Gateway) and the SCADA system level. Although Modbus TCP/IP as a protocol could support communication between field devices via TCP, i.e., between sensors, actuators and PLCs, at this point there is an additional requirement: The behavior as a real-time system (RTS). Real-Time support is a requirement mainly for devices at the field level (e.g., sensors and actuators). 

In an RTS, if the time constraints are not fulfilled, it can be said that the system has failed. As we have defined, the Modbus protocol TCP/IP implementation is in the application layer, therefore, taking into account the buffering problem, and given that the frames are queued FIFO (First Input First Output), unless a priority setting mechanism is used, as implemented by PROFINET RT, the process is not deterministic, and therefore is prone to introduce delay. In order to contextualize our work, Figure 2 shows an example of Modbus Network Architecture, where Modbus TCP/IP is not on the field device level, such as Modbus RTU (serial communication over RS-232, RS-485 and RS-422) or Modbus HDLC (MB+). Therefore, Modbus TCP/IP is suitable to enable the communication, for example, between two gateways, an HMI with a PLC, a gateway with a HMI, or an Input/output (I/O) device without RTS requirements. 

Therefore, our work focuses on level two in Figure 1 The high usability of the Modbus protocol at this level, in both Operational Technologies (OT) environments and within industry 4.0 or the Industrial Internet of Things (IIoT), (due to its ability to integrate into industrial processes such as process automation, industrial automation, building automation, power system automation and automatic meter reading), makes the security the main concern of the Modbus protocol. In that sense, we can affirm that the Modbus TCP/IP security problem has its focus on the protocol design. Modbus RTU was scaled to Modbus TCP/IP because the controllers could manage the bandwidth more efficiently, where a client (e.g., a SCADA) could support the connection with multiple servers; however, the Modbus frame TCP/IP was scaled without considering security.

The Figure 3 helps to understand the statement. As is shown, by default, Modbus PDU includes the Function Code field and Data payload. This function code indicates to the server which kind of action to perform. All function codes are found in the specifications [2]. When the Modbus TCP/IP frame was defined, only the Slave ID field was changed to an Application Protocol (MBAP) header, and the check error field was removed. MBAP contains only seven bytes [2]. Therefore, the frame does not include any mechanism to provide authentication or access control. In addition, the default Modbus specification [2] does not include a mechanism to provide integrity or confidentiality, using, for instance, end-to-end encryption. Therefore, if we adopt the criteria shown in [5], where it is established that design vulnerabilities are inherent to a protocol specification, present even in perfect implementations, we can confirm that Modbus security problems are related to the protocol design. 

However, until a few years ago it was not a problem, since industrial networks (traditional OT) were isolated. Now, it is the age of new factory (the industry 4.0) floor platforms, new technologies such as OPC-UA ([6] enhances the value of OPC-UA for industry 4.0), new paradigms, such as the Internet of Things (IoT) or IIoT, and the integration between IT and OT environments. Security is no longer a privilege, it is a necessity, and therefore mandatory.

Despite this, several vulnerabilities are found in devices that support the Modbus protocol, which are classified into vulnerabilities by protocol implementation (i.e., exploitation on a specific device because of, e.g., firmware error) or by protocol design (i.e., exploitation on any device using the protocol). 

For instance, in the CVE-2017-6819 vulnerability, during the communication between the operator and the PLC through the Modbus 0x5A function, it is possible for an attacker to send specially-crafted packets to consume the PLC resources, and hence freeze it. The product affected was the Modicon M340 PLC. The next reference details the Schneider Electric Report [7].

In addition, at the Defconf security conference in 2018, a study was presented where an injection attack was made upon three types of PLC (one Modicon, another Allen-Bradley and the third based on an open source PLC, known as OpenPLC [8]) which supported the Modbus protocol. To perform the injection attack, the same crafted frame was sent to each of the PLC, causing the same result: Denial-of-Service (DoS) [9].

In order to include security mechanisms to the protocol, in October 2018 the Modbus organization released security specifications [10], which provide robust protection through the blending of Transport Layer Security (TLS) [11] with the traditional Modbus protocol. TLS will encapsulate Modbus packets to provide both authentication and message-integrity protection. The security features leverage X.509v3 digital certificates for the authentication of both the server and the client. The protocol also supports the transmission of role-based access control (RBAC) information using an X.509v3 extension to authorize the request of the client.

Although the implementation of the system provides protocol security, authorized voices in the automation world argue that instead of securing Modbus, organizations should invest in technology to deploy a protocol that provides security by design, such as OPC-UA [12]. However, considering the number of devices supporting Modbus on the network, the option of providing security to the protocol is a solution that many organizations will adopt. 

Although the specifications are a guideline to provide security for the Modbus protocol, they have a general approach to implementation, leading to new proposals. For this reason the objective of our work is to improve the security of the Modbus protocol, based on the recommendations [10], i.e., our proposal is a way to contextualize the specifications [10] and to demonstrate the viability of it through both a security and a performance analysis. Within the implementation process, we introduce on the one hand the way in which the implementation is carried out, [, the authorization process, and on the other hand, the message authorization of the Modbus frame. Both points are not detailed in the specifications. To this end, this study proposes a role-based access control model (RBAC), which allows the server (e.g., PLC) to authorize a client (e.g., SCADA system) and that once this process has been carried out, and the Modbus frames flow through a secure channel, i.e., they are encrypted, they are also authorized. Since by default, TLS provides authentication of the server and client. Therefore, we are talking about an Authentication and Authorization (AA) model for the Modbus protocol. The authorization process is via a role-based access control. The roles are included as an arbitrary extension in the X.509v3 certificate and validated through a query from the server to a secure database, which has been populated by the client, e.g., the organization that own the SCADA, via an out-of-band (OOB) mechanism such as a secure web form. The authorization process takes place within the handshake phase of establishing a TLS connection, more precisely when the server receives the certificate from the client. Therefore, once this phase is over, any client-server communication will be secure, i.e., encrypted. At this point, the Modbus frame is also authorized, because the Modbus TPC frame contains a unique identifier (unit id) as part of its header, and also since this is transmitted in a secure way. So it can be used to authorize the frame, validated also via a query from the server to the secure database (the same used in the entity authorization process). In order to demonstrate the viability of our proposal, we provide both a security and performance analysis. The security analysis demonstrates the resistance of the proposal to different kinds of attacks. The performance analysis examines the latency behavior, not only for the cipher suites established by the security specifications ([10]), but also for others that involve a more complex processing and consequently higher latency measures. The remainder of the manuscript consists of related work in order to provide security to Modbus, followed by our implementation proposal. In addition, the corresponding security analysis is carried out. To evaluate our proposal, we will deploy our model on a GNS3 (Graphical Network Simulator-3) network built under the IEC-62443 standard that will allow traffic capture in a controlled environment. This represents our performance analysis. Finally, we provide the results obtained, the conclusions and the future research lines.

## 2. Related Work

Several efforts have been made to provide security to the Modbus protocol. In order to establish a balance around the analyzed proposals, they are divided between offensive security proposals and defensive security proposals.

On the offensive security side, the first works analyzed was the reference [13]. It presents a formal model for evaluating the security of the Modbus protocol based on a formal demonstration of the existence of man-in-the-middle (MiTM) attacks in Modbus-based systems. An additional work analyzed is the reference [14] which adopts a penetration testing approach using a penetration-testing tool based on Intrusion Detection Systems (IDS) to examine the insider threat, as well as the external threat through internal and external penetration testing, respectively. The work presented by the reference [15] involves an automated tool to generate malicious SCADA Modbus traffic to be used to evaluate such systems. Additionally, the work [16], demonstrates the attacks to the authentication protocol initially presented by [17]. A deep analysis of the Modbus protocol specification in order to distinguish the possible attacks was presented by [18]. The same work ([18]) identifies several taxonomies, divided into the serial transmission mode and Modbus TCP/IP protocol. All of them consider the existence of a Modbus sniffer or a packet injector. 

Other work analyzed was the reference [19]. It investigates the impact of malware attacks on Modbus-based SCADA networks, such as Code Red, Nimda, Slammer and Scalper. The authors also developed specialized malware to attack Modbus TCP/IP devices. One of them performs DoS attacks to the SCADA system by injecting valid but malicious Modbus packages, consuming bandwidth without alarming a possible IDS system that monitors the network.

On the defensive security side, i.e., mechanisms for detecting and preventing attacks, we have also been able to find numerous works. For instance, the article [20], proposes a special smart fuzzing technology for Modbus TCP/IP which satisfies the requirement of the vulnerability detection for Modbus TCP/IP. In addition, an abnormal traffic detection mechanism by tracing Modbus TCP/IP transactions is proposed by [21]. The proposed method ([21]) enables a response immediate and fast, not only to Denial-of-Service (DoS) attacks, but also to various types of malfunctions, such as routing loops, misconfigured devices and human mistakes. In addition, an authentication model, based upon the one-way property of cryptographic hash functions is proposed by [17]. Additionally, the article [22], investigates unauthorized, malicious and suspicious SCADA Modbus activities by leveraging the Darknet address space in order to establish attack prevention models. A solution based on SCTP (Stream Control Transmission Protocol) and HMAC (Hash-based Message Authentication Code) named ModbusSec, is presented by the reference [23]. The SCTP is a transport layer protocol that provides a reliable message-oriented communication channel, with features such as congestion control and multi-homing. A new secure version based on the TLS protocol which addresses some security problems of the Modbus is proposed by [24]. The experimental results show that it is feasible to implement TLS by using it as a benchmark of power grid applications. Finally, the work addressed by the reference [25] is the only precedent of a role-based access control (RBAC) system for Modbus. Additionally, this reference performs a detailed review of the root causes of vulnerabilities in industrial environments. The access control is done on the client side, since they developed a security-hardened architecture for delivering enhanced security for SCADA remote terminal (RTU) devices, i.e., not focused on Modbus TCP. However, it is an interesting proposal, because in addition to presenting the access control approach, they protect the frames cryptographically and check for the existence of a CRC, even though, it does not check for a valid CRC.

From the related work, we can conclude on the one hand that there is evidence of several efforts made, both offensive and defensive security, in order to provide security to the Modbus protocol in the TCP/IP version. In particular, there are two works closely related to our approach, [24], which proposes an implementation of TLS for Modbus TCP/IP, and [25], which proposes an RBAC model for Modbus RTU. However, these two schemes, and in general the rest of the efforts, are outside the context of the specification [10], which is the starting point of our proposal. On the other hand, we can affirm that our proposal presents novelty, because from the recommendations and guidelines provided by the specification [10], we propose a scheme that includes the following points. First, follows the recommendations of the specification, since it provides the implementation of an RBAC model, over a centralized system, which uses the X.509v3 certificate, for the server to authorize the client, i.e., the role of the client. All of the above procedure is performed within the framework of the TLSv1.3 handshake between the client and the server, which is not a condition established in the specification. Secondly, since the communication channel between the client and the server is protected after the handshake phase ends, i.e., encrypted, we perform the Modbus frame authorization from a unique identifier (unit id) found in the MBAP frame header (Figure 3). The second point of the proposal is not also addressed in the specification [10]. In order to demonstrate the viability of our proposal, our analysis includes two stages: Security analysis and performance analysis. The security analysis examines the resistance of the proposal to different kinds of attacks, such as eavesdropping, replay, forgery, and so on. The performance analysis verifies the implementation of the proposal, based on concrete and objective variables, which is measured with respect to changing conditions. These results are compared with established or adopted references. For our proposal, the variable is the latency; the changing conditions are the cypher suites, which, on one hand are defined by the specifications [10], and on the other hand, we propose to use other cipher suites which are more complex in terms of operations processing, i.e., resource consumption and the benchmarking are latencies and jitter of some industrial services.

## 3. Proposal of An RBAC Model on A Centralized Architecture

Our proposal consists of a Role-based Access Control (RBAC) model, which is based on a centralized architecture. Figure 4 shows the general architecture of the proposal. The general architecture of the proposal is formed by five sub-systems: The client, the MBAPS handler, the MBAP handler, the AC module and the Roles Database. It should be noted that the server is composed of three of the five sub-systems: The MBAPS handler, the MBAP handler and the AC module. MBAPS is the acronym for Modbus Application Protocol Secure. Below is a breakdown of how each of these entities interact as part of our proposal. The clients are sub-systems that send the connection request, e.g., a SCADA. Each client must store an X.509v3 certificate. The extension RoleSpecCertIdentifier has been added (Figure 4) to this certificate. We associate the OID ("1.3.6.1.4.50316.802.1") provided by the Modbus organization on the security specifications [10]. Additionally, our extension contains three fields. These fields are: roleName (e.g., operator), roleCertIssuer (e.g., client), roleCertSerialNumber (where we will store the Unit ID field). More details about the X.509v3 certificates can be found in the ITU recommendations [26]. The MBAPS is the entity or sub-system responsible for establishing the secure connection with the client, i.e., it is the entity that receives the client’s secure connection request, authenticates the client via certified, as part of the mutual-authentication process of TLS; hence, it also needs direct communication with the AC module (Figure 4). Once the secure connection has been established and the frame has been authorized, the MBAPS handler interacts with the MBAP handler to which it sends the Modbus frame. Therefore, this module participates in both the authorization and authentication processes. The AC module is in charge of executing the policies found performing the corresponding verifications in the role database (Figure 4). The trigger to perform its functions is received from the MBAPS handler module, so in addition to the role database, the AC Module only interacts with this entity (MBAPS Handler). 

The role database is a very simple entity whose interaction is based on one side with an out-of-band (OOB) mechanism through which the client populates the database before the connection request, and on the other side, with the AC module, which queries the stored data. This data enables one to perform access control policies (Figure 4). The MBAP handler module is the entity that communicates directly with the Modbus Memory Area (Figure 4) and for this, it must receive the Modbus frames from the MBAPS handler. Once the frames have been received, this module separates them according to their function code.

These entities interact as part of the RBAC model, which is composed of two authorization phases. The first uses the role extension of the client’s certificate to authorize it within the TLS handshake stage, and the second authorization phase enables each of the Modbus frames based on the unit_id. Additionally, as part of the TLS handshake, it executes an authentication process of the client and server entities. 

Therefore, the next section analyzes both the authorization and authentication phases as part of the handshake. In addition, the Section 3.2 analyzes the second authorization phase.

### 3.1. Authentication Phase via TLS and Entity Authorization Phase via Role on X.509v3 Certificate

As Figure 5 illustrates, the first step in the process is to populate the Role Database. We assume that there is an OOB mechanism (e.g., via a secure web form). The interaction of the role database with the OOB system is shown in Figure 4. Through this system, the client will be able to insert the Role that they will have in the server, as well as Unit ID that the Modbus frames will have in their header. The client then sends a request to establish a secure connection to the server. Until step 8 in Figure 5 a normal handshake, this process is performed between a client and a server as part of a TLS implementation. However, this normal handshake includes an important feature of TLS, the first step of the mutual authentication, i.e., the server authentication through the verification of the server certificate (step 5). In step 9, the client adds an extension with the corresponding *role* to its certificate, as mentioned in the previous section, before sharing it with the server. All of this as part of the TLS session management. 

Once, the MBAPS handler has received the X.509v3 certificate (step 10), it verifies the certified (step 11). This process is the second step of the mutual authentication process, i.e., client authentication (step 11). In addition, the server extracts from it the *role*; sends it to the AC module, which queries the Roles Database and executes the RBAC policy (step 12 A–step 12 D). This RBAC policy validates that the *role* stored in the database coincides with the one included in the certificate. Without knowing if their role will be authorized, the client sends relevant information to the server (steps 13–15). These steps are also common within the TLS session generation process. Until the server receives the “Finished” message from the client (step 15), the MBAPS handler will retain the authorization or non-authorization response received from the AC Module. If the role is authorized, the MBAPS handler sends the messages to confirm the cipher specifications and finishes, which steps are also regular in the generation of a TLS session. If the role has not been authorized, the handshake phase ends when the MBAPS handler sends an exception error message to the client and closes the connection.

### 3.2. Message Authorization Phase

Once the authorization phase between client and server is over, frames can be exchanged securely, using symmetric encryption. These frames will contain the traditional Modbus frames (MBAP + PDU (Figure 3)). In the Figure 5, when the MBAPS Handler receives the frame request (MBAP + PDU) (step 18), it extracts the unit_id from the MBAP header and sends it to the AC module (step 19), which queries the Role Database (steps 20–21) and executes the authorization method. The authorization method validates that the unit_id stored in the database matches with the one included in the MBAP header field (step 22). Then the AC module responds by authorizing (or not) the frame (step 23), from which MBAPS handler sends an exception, error message to the client (step 24.B1) or sends the frame to the MBAP handler (step 24.A1) that processes the frame from the function code and interacts with the four areas of the Modbus Memory Area (MMA). The Figure 4 shows the four areas of the MMA.

## 4. Implementation Phase

From the sequence diagram of the Figure 5, in the Figure 6 we divided into logical actions these design phases in order to simplify the implementation phase. There are three challenges at this point: (1) Select the base library (core library) on which to implement, (2) analyze the mechanism to implement encryption and to enable other security features and (3) analyze the cipher suites to implement. 

Our Zenodo reference contains the implementation details that include the Role Database, the arbitrary extension configuration file and the video demonstration [27].

### 4.1. Selection of the Core Library

In order to accomplish our implementation we used the open source library, supported by the Community pymodbus [28]. Although as part of the evaluation phase is shown through performance analysis details of latencies between client and server, without any encryption, i.e., using the unmodified pymodbus library, at this point we can mention that these measurements of latencies was the first criterion used for the selection of the library (see column TCP of both tables of Section 6.1). The second criterion considered was the language in which it was written: Python; and in third place was analyzed the broad community that has pymodbus, hence, the troubleshooting is simplest.

### 4.2. Mechanism to Implement the Encription and Other Security Requirements

Once the core library was selected, we proceeded to implement each of the steps shown in Figure 6, where we show the followed server-side procedure. In order to generate the handshake session a TLS socket was used. Over this socket, the client sends the certificate with the added role extension. Next, the role of the client’s certificate is extracted, the database is queried to recover the associated role to the specific OID, the policy is executed, i.e., it is checked that the role stored in the database, as well as the extracted role from the client certificate, do in fact match, determining the client’s authorization. At this point, the handshake stage and the asymmetric encryption stage are completed. 

Therefore, when a Modbus frame is received, the ID unit is extracted from the header, the database is queried again, and the authorization method is executed, i.e., the unit id associated with the OID of the certificate of the client is verified to match with the frame. Finally, the frame authorization is determined, allowing the core library to perform the function code over the Modbus Memory Area (MMA).

Next we detail the three most important points during the implementation stage: (1) The library used to generate the TLS socket, (2) the certificate generation process and (3) the database where the roles are stored.

The implementation of TLS was based on the recommendations provided by Python Software Foundation (PSF) from the wrap_socket handler [29]. This module provides access to TLS (a.k.a., “Secure Sockets Layer”) encryption and peer authentication facilities for network sockets, both client-side and server-side. This module uses the OpenSSL library. This library supports the use of TLSv1.3, and the management of X.509v3 certificates, where both are essential requirements in our implementation. Another advantage of the library is that it presents the SSLContext.set_ciphers() method, which enables the efficient management of the cipher suites to be selected. The SSL module disables certain weak ciphers by default, but it is possible to restrict the choice of ciphers even further. The following subsection analyzes the used encryption suites.

In order to generate an arbitrary extension, i.e., an extension with a custom OID, a configuration file (openssl.conf) is previously generated, and it is loaded in the command line to generate the certificate. The expression (1) shows the command used to generate the certificate with the added extension. This extension is contained in the openssl.conf configuration file. For more detail, our Zenodo reference contains these implementation details.

Finally, the Role DB database has been implemented using SQLite database. For instance, expression (2) is the simple instruction used to create the Table over our “RoleDB.db” to perform the proof of concept (PoC). It is composed of a table with the attributes: Role, oid, and unit_id, which are gotten from the same client, and are used to authorize the client and to authorize the Modbus frame, respectively.

openssl req -newkey rsa:2048 -nodes -keyout {0} -extensions RoleSpecCertIdentifier -out {1} -subj “/C=NA/ST=NA/L=NA/O=OT/CN={2}/description={3}” -config ./openssl.cnf,(1)

CREATE TABLE roleTable (ID INTEGER PRIMARY KEY, unit_id INTEGER, role text, oid STRING),(2)

### 4.3. Cipher Suites to Implement

The specifications [10] set the lowest boundary of the cipher suites to be used, in terms of resource consumption due to processing, while maintaining minimum security levels. However, as cipher suites will be used as a resource in our evaluation phase for performance analysis, we have decided to perform an analysis of other cipher suites and to establish a comparative analysis with respect to those defined by the specification [10]. The cipher suite represents which cryptographic algorithms and methods should be used, and it is defined by [30]. Currently, there are 339 suites officially supported, that target different applications and security levels. The cipher suite name contains the key exchange, the authentication method, the key exchange algorithm and the symmetric algorithm for an authenticated encryption of application data transfer between entities.

For instance, TLS_ECDH_ECDSA_WITH_AES_128_CBC_SHA (0xC004) states that the Elliptic-curve Diffie-Hellman (ECDH) will be used as key exchange, while Elliptic Curve Digital Signature Algorithm (ECDSA) will be used as digital signature, and AES_CBC_SHA will be used to construct the symmetric authenticated encryption (Advanced Encryption Standard (AES, a.k.a. Rijndael) with Cipher Block Chaining (CBC) mode for encryption and HMAC with SHA for the construction of the Message Authentication Code). Table 1 sets the encryption suites recommended by the specifications [10] and Table 2 sets the cipher suites that are additionally sampled.

As it can be noticed when comparing both tables, the cipher suites proposed in Table 2 are more complex than those proposed in Table 1, i.e., in terms of the computational cost required to implement each. However, in order to select the suites that we have proposed in Table 2, we have examined the corresponding relationship with those presented in Table 1, with the objective of stressing the system and obtaining measures of higher latencies. For instance, if we compare 0xC02B with 0xC02C, a similar structure is visible in terms of the key exchange, the authentication method and the symmetric algorithm construction for encryption, however, on the one hand, 0xC02B uses AES_128_GCM while 0xC02C uses AES_256_GCM, and on the other hand, 0xC02B uses SHA256 while 0xC02C uses SHA384.

## 5. Evaluation Phase

At this point, we consider it important to analyze the phases of the proposal in order to understand the evaluation phase. The Figure 7 contains the roadmap followed. Once the design and implementation phase has been completed, we will divide the evaluation phase into two stages. The first, corresponding to the next section, exposes our proposal to a security analysis, while the second corresponds to the performance analysis of the proposal. The purpose of the evaluation phase is to determine whether the proposal is feasible.

### 5.1. Security Analysis

The security analysis of a system traditionally starts with the definition of the attacker’s model. Since we do not strictly propose a protocol, but rather we base our analysis upon the improvements proposed by the specifications [10], we have divided our security analysis into some issues of an attacker model, as well as the analysis with respect to other parameters and the pillars of cybersecurity. At this point we should mention that in our demonstration the database is local. However, in the specification [10], it is indicated that it can be external to the server, so in that case it would be secured via a TLS socket. The attacker model used is the classic Dolev-Yao [31]. The following analysis is a consequence of the analysis of the exchange of messages between a client and a server through the network.

Mutual Entity Authentication: Our proposal contains one mechanism where authentication is implemented. It is the TLS authentication proper, which happens when the endpoints verify the validity of the certificates. These certificates has been previously installed, i.e., they are factory-installed.Confidentiality and Message Authentication: As the recommendations [10] indicate, providing security to MBAP through the implementation of TLS via the construction of symmetric encryption, i.e., encryption + MAC, a.k.a., authenticated encryption (e.g., in Table 1, AES 128 CBC will be used for encryption) ensures the confidentiality of the information.Integrity: TLS provides a security-focused protocol alternative to MBAP (to see MBAP sub-system in Figure 4) by adding data integrity via certificates, and authorization via information embedded in the certificate, such as user and device roles. So, the integrity is provided by on one hand the public-key cryptography in the TLS handshake process, and on the other hand by symmetric encryption (encryption + MAC).Replay attack: TLS properties guarantee freshness, so TLS protects against replay attacks. In addition, an attacker cannot replay a message that has been logged in previous sessions, because both the oid and the unit id change in every session. Moreover, it must be taken into account that the generation of the oid, the role and the unit id is performed through an OOB mechanism (Figure 4)Man-in-the-Middle (MiTM) attacks: Because all of our client-server communications are encrypted through TLS, mutual entity authentication is required before performing a transaction. Mutual entity authentication exists, because according to TLS, it requires that each end-point send its domain certificate chain to the remote end-point (step 5 and step 11 of Figure 5) Subsequent communication between entities over authenticated encryption (symmetric encryption algorithms) also provides a MAC algorithm, which protects the communication against MiTM attacks.Eavesdropping attack: Thanks to the implementation of TLS, our proposal guarantees the confidentiality and integrity of the information. For this reason, we consider our proposal resistant to any eavesdropping attack.

#### Implementation Test

Summarizing, the demonstration of resistance to the attacks mentioned above, as well as the guarantee that the pillars of cybersecurity are not compromised, is because TLS, by default, provides security to MBAP by adding data confidentiality, data integrity, anti-replay protection, end-point authentication via certificates and authorization via information embedded in the certificate, such as user and device roles. Additionally, we provide a mechanism to authenticate the Modbus frame.

Therefore, from a design point of view, we can justify our analysis, which means that if there was a failure it would be due to the implementation. For this reason, we carry out a new demonstration by using a framework to measure the quality of the software.

For Python, there is a well-known library called Pytest. Pytest automatically catches warnings during test execution. In addition, pytest can be synchronized with Allure. Allure Framework is a flexible lightweight multi-language test report tool that not only shows a very concise representation of what has been tested in a neat web report form, but also allows everyone participating in the development process to extract a maximum of useful information from the everyday execution of tests. Therefore, Allure-Pytest is our Software Quality Framework.

Figure 8 contains the most basic part of the test we apply to TLS, where (the client’s ip that appears in Figure 8) matches the client’s ip of the evaluation architecture, which will be analyzed in the Section 5.2.1. Multiple tests have been carried out, including tests for connection and disconnection, data transmission and data receipt. The Figure 9 shows a part of the report that is generated through Allure, where it is possible to monitor the number of tests performed and their success rate. Although there are more rigorous frameworks, Pytest gives a good measure of the correct implementation of our security proposal.

### 5.2. Performance Analysis

The performance analysis is given by the deployment of an evaluation architecture, from a test-network, which is built on GNS3 and is based on the industrial standard IEC-62443, which divides the network into levels. Both the client and the server are deployed on this architecture in order to measure the latencies generated for different cipher suites. The results of these measurements are compared with the time constraints collected for several industrial services. Therefore, each of the sections below are presented according to the logical order mentioned, i.e., (1) evaluation architecture, (2) tool for measuring the latency, (3) test scenarios to be executed and (4) time constraints in industrial services.

#### 5.2.1. Evaluation Architecture

In order to evaluate our proposals, models that use an in-depth defense approach are used, aligned with industrial security standards such as IEC-62443 ICS Security and NIST 800-82 Industrial Control System (ICS) security [32]. IEC-62443 divides the network of an industrial environment into levels. Based on the application environment defined in Section 1 we consider that our implementation should be based upon levels one and two of the Purdue Model. Level 1 contains all of the controlling equipment. The main purpose of the devices (e.g., our server) in this level is to open valves, move actuators, start motors, and so on. Typically, in Level 1 we find devices such as PLCs, Variable Frequency Drives (VFDs), dedicated proportional-integral-derivative (PID) controllers, and so on [33]. In addition, Level 2 specifies parts of the system to be monitored and managed with HMI systems (e.g., our client), which allows to start or stop the machine and see some basic running values and manipulate machine specific thresholds and set points [33]. 

According to this requirement we generate a testnet in GNS3, which includes a firewall (ASA 5505), switches L2 (cisco 2960), routers (cisco 1941) and Docker containers to simulate the Modbus TLS Client and the Modbus TLS Server. The Cisco Adaptive Security Appliance (ASA) is an advanced network security device that integrates a stateful firewall, VPN, and other capabilities. This testnet employs an ASA 5505 to create a firewall and protect an internal industrial network from external intruders, while allowing internal hosts’ access to the Internet. 

The ASA creates three security interfaces: Outside, Inside, and DMZ. It provides Outside users limited access to the DMZ, and no access to inside resources. Inside users can access the DMZ and outside resources. For this reason, the Modbus TLS client has access to the Modbus TLS server.

Testnet configurations include NAT, VLAN and Access Control over the ASA 5505. We share all configurations (ASA 5505, R1, R2, and R3) as an additional resource using a Zenodo reference [27]. The architecture designed is shown in the Figure 10.

#### 5.2.2. Tool for Measuring Latencies

Once the architecture is deployed, and taking into account that our development was based on python, we use the *time* module. The python docs say that clock should be used for benchmarking. In the time module, there are two timing functions: These are *time* and *clock*. In expression (4), the variable t_f_ is elapsed CPU seconds since t_0_ was started (expression (3)). By moving the start and end within the code we can obtain the results of interest. We have recorded the latency times for the client, i.e., when the client requests a function (e.g., read coils), this function is executed in the server and until the response is received.

t_0_ = time.clock(),(3)

t_f_ = time.clock() - t_0_,(4)

#### 5.2.3. Test Scenarios for Performance Evaluation

On the one hand, the Modbus specifications [2] list the set of functions that are implemented, and additionally describes therefore the specific functions to be tested in the PI-MBUS-300 guidelines [34]. These three specific functions should be executed over the maximum number of coil (65535) or register (123) allowed by the protocol. These three functions are: Read Coils (0x01), Read Holding Registers (0x03) and Read-Write Multiple Register (0x17).

On the other hand, in order to evaluate the contribution of the RBAC model to the latencies, the following two approaches have been tested. First, Modbus over TLS, but without the RBAC model, is used to perform all of the measurements. This means that some steps that were included in Figure 6 are skipped for this first approach. More specifically, in the first phase of the extraction of roles from the certificate, the database query and the execution of the RBAC policy are removed, while in the second phase the extraction of the unit ID, the database query, the authorization mechanism and the authorization response, are removed. These results will be analyzed in Section 6.1.

In the second approach to be tested, all the steps also including the RBAC model (described in Figure 6) are executed at each test. These results will be analyzed in Section 6.2, and as it has more steps than the measurements without RBAC policy (approach 1), it is expected that obtained latency measurements will be higher in the second approach.

In both approaches, there are several options for selecting the cipher suite to be used, as established in the specifications [10] (see Table 1 and Table 2). Hence, all of these cipher suites will also be tested in order to analyze the contribution to the final latencies in both approaches. Additionally, as we are based on Modbus TCP and it is non deterministic, each test will be repeated 1,000 times. The Table 3 summarizes all the tests that have been carried out.

As an example, the implementation used to evaluate the performance using the Read Holding function to the TCP samples is shown in the Figure 11 In order to execute the function on TLS, instead of using the ModbusTcpClient function, the ModbusTLSClient function is used. The cycle’s variable is in charge of monitoring the number of samples that are collected (to consider that the client’s ip, Figure 11, also matches the client’s ip of the evaluation architecture, Figure 10).

Finally, it is necessary to specify the measurement process with regard to the TLS session and the number of times that a function will be executed per session, since this is a requirement that we have established. As we mentioned before, the Figure 6 illustrates how for a TLS session, the process by which a single function (e.g., read coil (0x01)) is executed once, is composed of two phases. A phase where the TLS session is established (asymmetric encryption, Figure 6) and another where the Modbus frame is authorized, as well as the interaction between the MBAP handler and the MMA, i.e., the execution of the read coil function (0x01), under symmetric encryption, i.e., encryption + HMAC (Figure 6). Although it would be possible to execute many functions within the same TLS session, in the tests only one function is executed: In order to obtain the worst case latency, we have generated a new session for each one of the functions to be executed. 

#### 5.2.4. Latencies Benchmarking

Once the measurements are carried out, we set latency constraints in order to reach conclusions about proposals feasibility. In order to achieve this, we take as a reference an ITU appendix: “Technical and operational aspects of Internet of Things and Machine-to-Machine applications by systems in the Mobile Service” [35]. Table 4 shows the latencies and jitter of some industrial services. ITU defines latency such as a “parameter for characterizing the communication service delay from an application point of view”. In addition, it defines jitter, such as “variation of latency”.

It is necessary to emphasize firstly that most of the services listed in Table 4 are deterministic in nature, and as we have mentioned in the introduction section, our work focuses on Modbus TCP/IP, which is non-deterministic. However, we will adopt these values as latency constraints to be fulfilled. Secondly, it is important to remark that Modbus can be used in more application areas (see Table 5) than those identified in Table 4. Our proposal will be feasible if it finally fulfills the latency constraints defined for the specific use case, which might be even higher that those presented in Table 4.

## 6. Results

Although security analysis has determined that our proposal is resistant to different attacks and that pillars of cybersecurity have not been compromised, it is also relevant to analyze its performance in order to assure the feasibility of our proposal (see Figure 7). For this reason, this section discusses the results of the tests described in Section 5.2.3. 

As a reminder, the first section includes the performance analysis for the tests carried out when using different cipher suites but without applying RBAC model, for the different Modbus functions.

The second section includes the performance analysis for the tests carried out when applying RBAC model, comparing also results for different cipher suites and different Modbus functions. 

In both approaches, each test was repeated 1000 times.

### 6.1. Comparison between Cipher Suites Latencies without Applying RBAC Model

Table 6 shows the average and standard deviation of the 1000 measurements of latencies obtained for each combination of cipher suite and Modbus function.

As shown in Table 6 the fastest tested null-encryption suite with the secure hash function, TLS-RSA-WITH-NULL_SHA256 (0x003B), is in average 0.09 ms (Read Coils), 0.14 ms (Read Holding Registers) and 0.18 ms (Read-Write multiple Registers) faster than the lower latency encrypted option (0x0035) which implements TLS-RSA-WITH-AES_256_CBC_SHA256. In addition, the average latency of the former cipher suite (0x003B), is 0.44 ms (Read Coils), 0.43 ms (Read Holding Registers) and 0.55 ms (Read-Write multiple Registers) slower than the insecure Modbus (Modbus TCP). This shows that the highest latency is provided by asymmetric encryption implemented by the TLS handshake.

Although it is difficult to establish a comparison between CBC and Galois/Counter Mode (GCM) with these data, given that our approach depends on the requirements of the specification [10], GCM beats CBC categorically. For instance, Table 6 contains the encryption suite (0x003D), while Table 7 contains the encryption suite (0x009D). Clearly, it is observed that despite presenting a more complex hash function the latencies of 0x009D are lower. Table 7 shows that despite the high requirements (AES_256) and (SHA384), TLS_ECDHE_ECDSA_WITH_AES_256_GCM_SHA384 would be faster than either of those, use a lot less bandwidth, and be more secure.

Standard deviation is also calculated, taking into account that the process that handled the communication is not deterministic. If it is taken into consideration that a low standard deviation indicates that the data points tend to be close to the mean of the set, while a high standard deviation indicates that the data points are spread out over a wider range of values. From Table 6 and Table 7 we conclude that as the operations complexity tends to increase, the standard deviation increases and therefore the latency values tend to be further away from the mean. For instance, if the cipher suite 0x009C is compared with 0x009D, both share operations, but the operations from 0x009D are more complex. This situation is reflected also in each value of the standard deviation.

The Figure 12 confirms the analysis performed, once the maximum, minimum and mean values of the latencies associated with each of the three test functions have been recovered. As both the processing of the operations of the encryption suites, as well as the processing of the Modbus function increase, the latency values also increase. If we compare the three functions associated with 0x003C, the greatest increase in latencies is given through the "read-write multiple records" function. Additionally, if we compare 0x003B without symmetric encryption and 0x003C with AES_CBC_128 for the same function, e.g., "read-write multiple records", we observe that latencies increase due to the processing introduced by symmetric encryption.

Despite the additional computational complexity, even the slowest cipher suite achieves transaction times below to those shown in Table 4 related to Factory automation (not for motion control), Process automation (remote control and monitoring), Electricity distribution (medium voltage) and intelligent transport systems. Anyhow, encryption algorithms with block cipher operation, such as AES-128-CBC, show the worst performance. Therefore, stream operated ones should be preferred, such as AES-256-GCM, for high parallelizable devices.

### 6.2. RBAC Model Results

Table 8 shows the average and standard deviation of the 1,000 measurements of latencies obtained for each combination of cipher suite and Modbus function, where at each execution of the test the entire process of the RBAC model was carried out as described in the Figure 6.

The increase of the latencies in Table 8 with respect to the latencies analyzed in Table 6 and Table 7 is coherent. On one hand, because more steps are carried out: When the RBAC model is executed, in addition to the TLS handshake and the symmetric encryption, in the server side the role is extracted from the certificate, two queries to Role database are executed and two policies are verified. On the other hand, the client must generate the certificate extension. All these additional steps imply extra processing.

A conclusion from Table 8 is the implication of the execution of Modbus functions on latency. For instance, for all cases it is fulfilled that the difference between the average times of the reading operations is less than the difference between the average times of either of the two reading operations with respect to the read-write operations.

With respect to the standard deviation, the conclusion reached in the previous section is verified, where it was determined that the standard deviation increased depending on the complexity of the operations to be performed. This increase occurs both when the complexity of the Modbus function increases, as well as when the complexity of the cipher suite increases. An analysis of the results in Table 8 for the cipher suites compared in the previous section shows that the behavior is the same. These conclusions can be clearly seen in the Figure 13. For instance, it is evident from 0x009C, the latencies generated by “read-write multiple registers” is higher than “read coils”. In addition, if we compare the behaviors of the 0x009D cipher suite with the 0x003D cipher suite, for the same function, e.g., "read-write multiple registers", we can see that the latency generated by 0x009D is higher, which is associated with a higher hash function despite using GCM, while 0x003D uses CBC.

Although latency increases with respect to the values analyzed in Table 6 and Table 7, it is demonstrated that our proposal is feasible to be used in remote control and monitoring in Process automation, as it fulfills with the latency constraints shown in Table 4. However, at this point it must be reminded that all processes included in Table 4 are deterministic, which is not a requirement for our application context. Finally, it is important to highlight that these results were for the worst case scenario, where for each execution of the Modbus function a new session was established. 

## 7. Conclusions

Applications that use SCADA systems rely upon protocols such as Modbus to enable their operations. Many protocols widely deployed lack basic security mechanisms, such as confidentially and the authenticity of transmitted data. When deployed in critical infrastructure assets, these applications enable advanced control possibilities. The exposition of those systems by incorrect deployment or existing vulnerabilities, both in design and implementation, create new attack scenarios. Based on the security recommendations established by the Modbus organization, our manuscript includes a role-based access control model (RBAC) as an access control mechanism, in order to authorize and authenticate systems based on Modbus. This model is divided into an authorization process and an authentication process. The authentication process is provided by TLS, because by default this implements mutual authentication. The authorization process includes both the entity authorization as well as the message authorization. The entity authorization is via roles, which are included as an arbitrary extension in the X.509v3 certificate. The roles are validated with a value stored in a secure database, populated from an out-of-band mechanism. The message authorization process is via a unit id, where it comes from a unique identifier containing the frames, which is validated from a query to the secure database (same database used by the authorization process). In order to evaluate our implementation that we have based on a security analysis, which indicates the attacks against which our implementation is resistant, also justifying the correct implementation. In addition to the security analysis, we perform a performance analysis, since the cipher suites support the mechanism mentioned above different cipher suites were analyzed, establishing a comparison with benchmarks, arriving at the conclusion of the feasibility of the model presented.

Since one of our lines of research is access control, based on attributes (ABAC) in decentralized systems ([40,41]) and this model of access control based on roles (RBAC) has been applied in a centralized environment (Role Database), the first future line of research is to apply this model in a decentralized environment based on blockchain. Additionally, we want to provide results to systems based on both Hyperledger Fabric Blockchain and Ethereum blockchain. Thirdly, the creation of a robust mutual authentication RFID protocol that works together with our ABAC blockchain system in order to build a secure supply chain system.

## Figures and Tables

**Figure 1 sensors-19-04455-f001:**
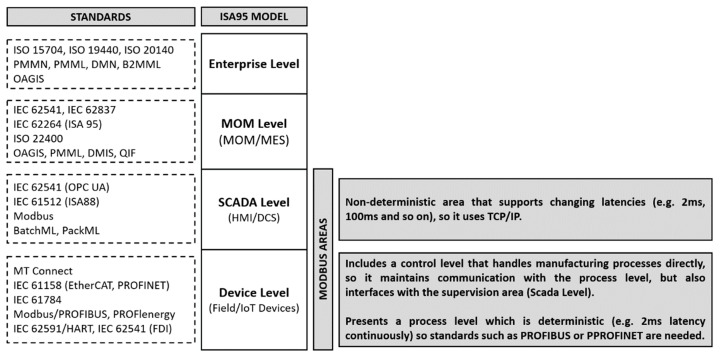
Action fields of Modbus based on the ISA 95 model and related standards [3,4].

**Figure 2 sensors-19-04455-f002:**
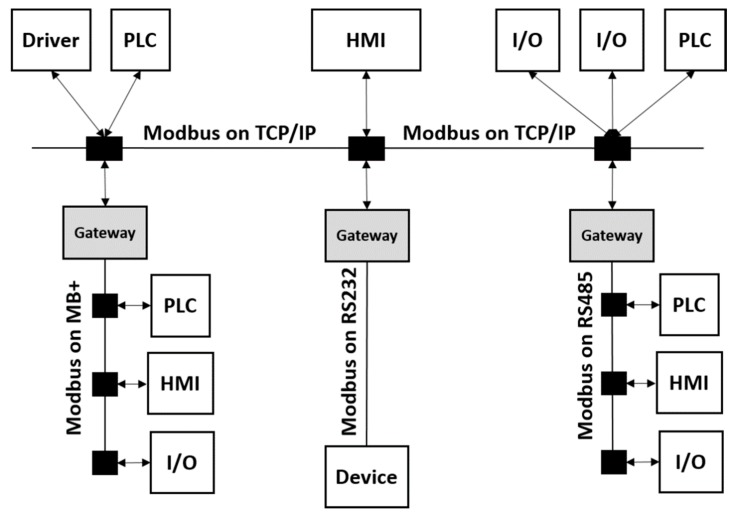
Example of Modbus Network Architecture [2].

**Figure 3 sensors-19-04455-f003:**
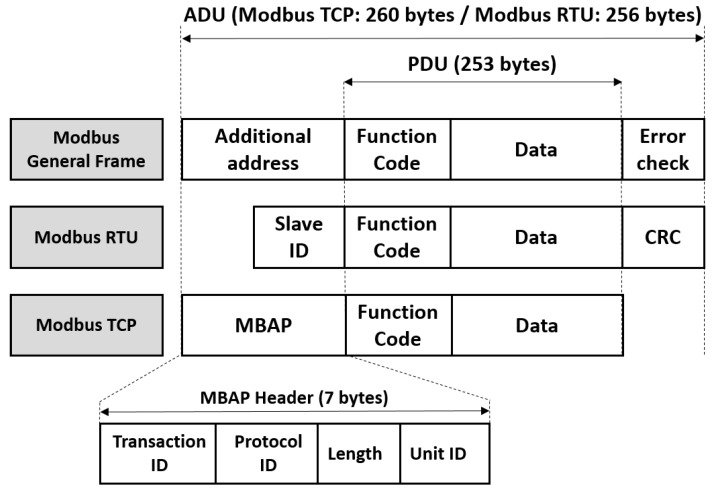
The Modbus frame [2].

**Figure 4 sensors-19-04455-f004:**
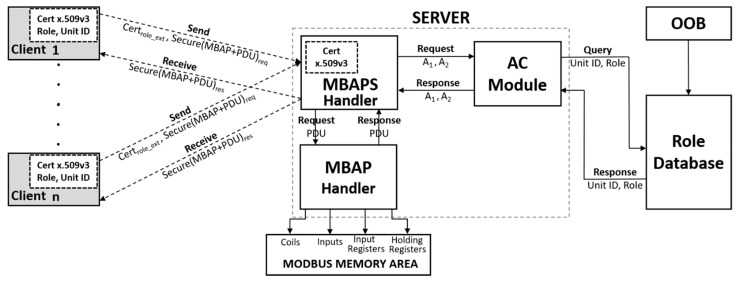
Role-based access control model (RBAC) based on centralized architecture.

**Figure 5 sensors-19-04455-f005:**
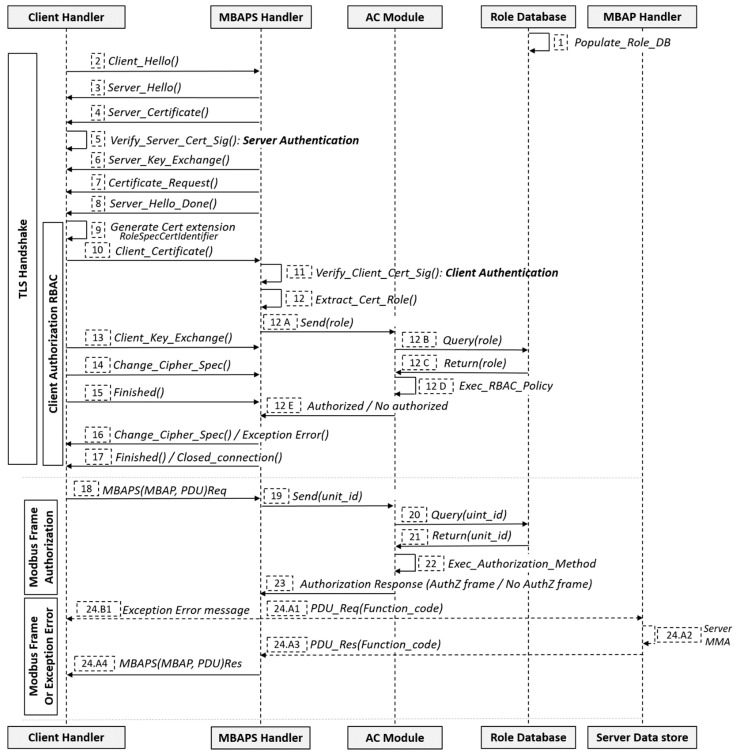
Sequence Diagram of Modbus Transport Layer Security (TLS) handshake and RBAC client authorization.

**Figure 6 sensors-19-04455-f006:**
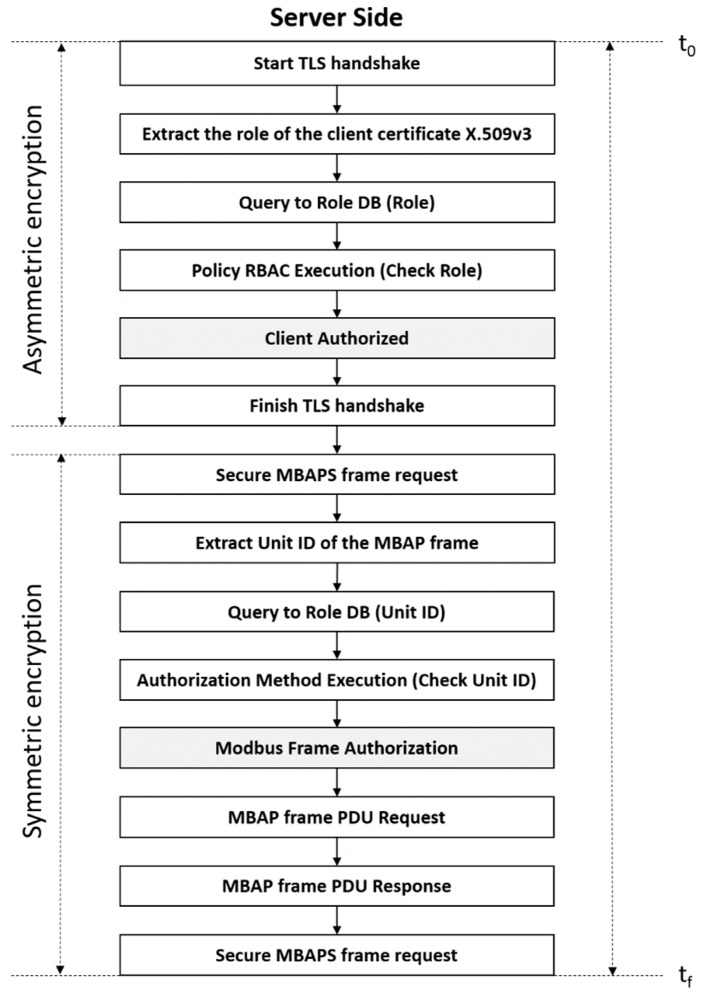
Step-by-step, successful server-side; the two-authorization process.

**Figure 7 sensors-19-04455-f007:**
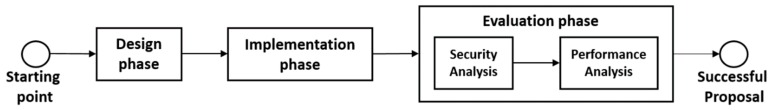
Proposal roadmap.

**Figure 8 sensors-19-04455-f008:**
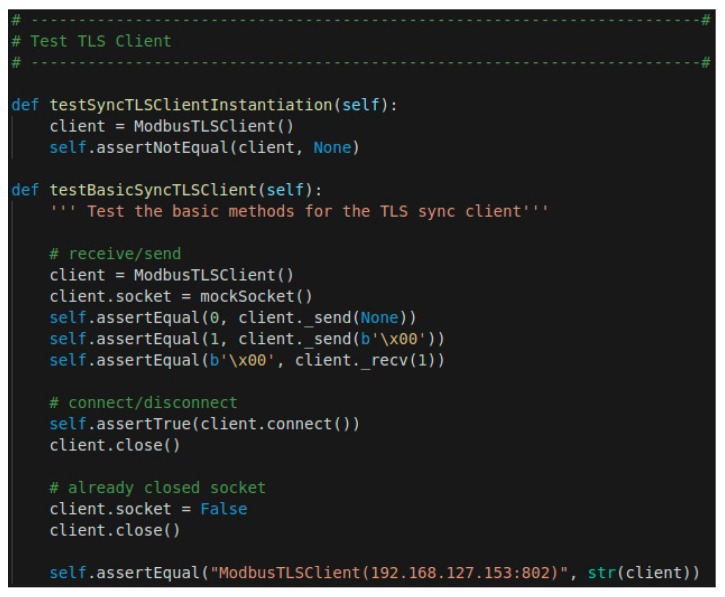
Test TLS client via the Pytest module.

**Figure 9 sensors-19-04455-f009:**
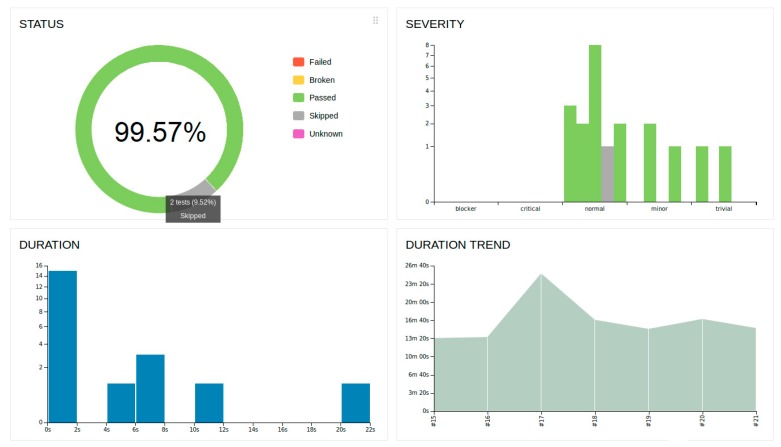
Report sample generated through Allure.

**Figure 10 sensors-19-04455-f010:**
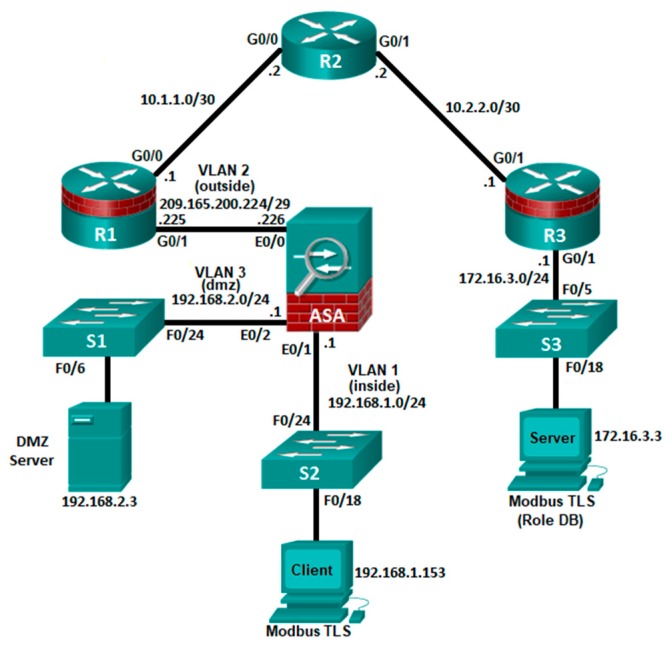
Diagram of the architecture deployed on GNS3.

**Figure 11 sensors-19-04455-f011:**
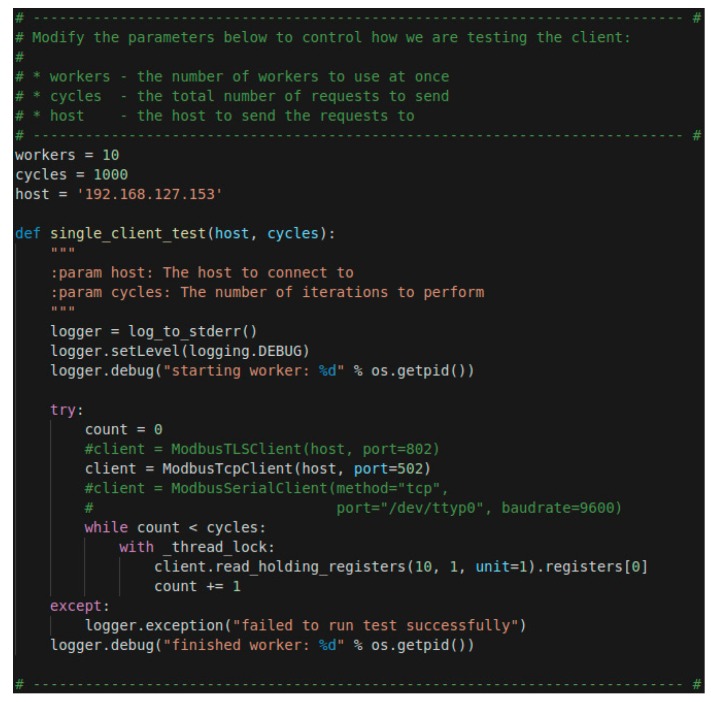
Read Holding function used to evaluate the performance.

**Figure 12 sensors-19-04455-f012:**
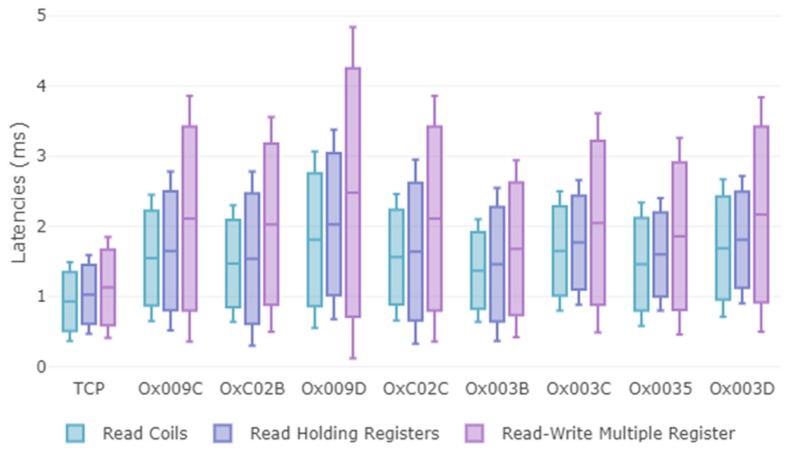
Boxplot (maximum, average, minimum) of latencies (ms) for each of the cipher suites (without applying RBAC model).

**Figure 13 sensors-19-04455-f013:**
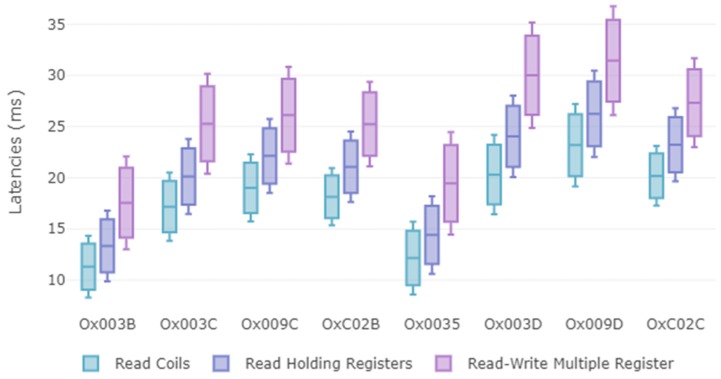
Boxplot (maximum, average, minimum) of latencies (ms) for each of the cipher suites when applying RBAC model.

**Table 1 sensors-19-04455-t001:** Cipher suites defined by the specification [10].

Cipher Mode	Cipher Suite	Number ^1^
Null	TLS_RSA_WITH_NULL_SHA256	0x003B
CBC	TLS_RSA_WITH_AES_128_CBC_SHA256	0x003C
GCM	TLS_RSA_WITH_AES_128_GCM_SHA256	0x009C
	TLS_ECDHE_ECDSA_WITH_AES_128_GCM_SHA256	0xC02B

^1^ IANA Format (0xC0, 0x2D) equal to Number (0xC02D).

**Table 2 sensors-19-04455-t002:** Cipher suites used as additional samples.

Cipher Mode	Cipher Suite	Number ^1^
CBC	TLS_RSA_WITH_AES_256_CBC_SHA	0x0035
	TLS_RSA_WITH_AES_256_CBC_SHA256	0x003D
GCM	TLS_RSA_WITH_AES_256_GCM_SHA384	0x009D
	TLS_ECDHE_ECDSA_WITH_AES_256_GCM_SHA384	0xC02C

^1^ IANA Format (0xC0, 0x2D) equal to Number (0xC02D).

**Table 3 sensors-19-04455-t003:** Total samples analyzed.

Tested Approach	Tested Cipher Suites	Tested Modbus Functions	Number of Tests
Without RBAC	9	3	27000
With RBAC	8	3	24000
		Total:	51000

**Table 4 sensors-19-04455-t004:** End-to-end latency constraints [35].

Service	End-to-End Latency	Jitter
Factory automation (motion control)	1 ms	1 µs
Factory automation	10 ms	100 µs
Process automation (remote control)	50 ms	20 ms
Process automation (monitoring)	50 ms	20 ms
Electricity distribution (medium voltage)	25 ms	10 ms
Electricity distribution (high voltage)	5 ms	1 ms
Intelligent transport systems (infrastructure backhaul)	10 ms	2 ms
Remote Control	5 ms	1 ms

**Table 5 sensors-19-04455-t005:** Modbus use cases.

Service	Reference
Process automation	[36]
Industrial automation	[34]
Building automation	[37]
Power System Automation	[38]
Automatic Meter Reading	[39]

**Table 6 sensors-19-04455-t006:** Average latencies (ms) and standard deviation in Cipher Block Chaining (CBC) cipher mode (without RBAC model).

	Read Coils (0x01)		Read Holding Registers (0x03)		Read-Write Multiple Register (0x17)	
Cipher Suite Code	Ave.	Std.	Ave.	Std.	Ave.	Std.
TCP	0.93	0.56	1.03	0.61	1.13	0.72
0x003B	1.37	0.73	1.46	1.09	1.68	1.26
0x003C	1.65	0.83	1.77	0.89	2.05	1.56
0x0035	1.46	0.80	1.60	0.87	1.86	1.43
0x003D	1.69	0.91	1.81	0.98	2.17	1.67

**Table 7 sensors-19-04455-t007:** Average latencies (ms) and standard deviation in Galois/Counter Mode (GCM) cipher mode (without RBAC model).

	Read Coils (0x01)		Read Holding Registers (0x03)		Read-Write Multiple Register (0x17)	
Cipher Suite Code	Ave.	Std.	Ave.	Std.	Ave.	Std.
TCP	0.93	0.56	1.03	0.61	1.13	0.72
0x009C	1.55	0.91	1.65	1.13	2.11	1.75
0xC02B	1.47	0.83	1.54	1.24	2.03	1.53
0x009D	1.81	1.26	2.03	1.35	2.48	2.36
0xC02C	1.56	0.93	1.64	1.31	2.11	1.77

**Table 8 sensors-19-04455-t008:** Average latencies (ms) and standard deviation for each of the cipher suites when applying the RBAC model.

	Read Coils (0x01)		Read Holding Registers (0x03)		Read-Write Multiple Register (0x17)	
Cipher Suite Code	Ave.	Std.	Ave.	Std.	Ave.	Std.
0x003B	11.31	3.01	13.34	3.45	17.55	4.54
0x003C	17.17	3.34	20.12	3.65	25.27	4.89
0x009C	19.01	3.29	22.13	3.61	26.11	4.73
0xC02B	18.14	2.79	21.07	3.43	25.24	4.12
0x0035	12.15	3.55	14.41	3.79	19.45	5.01
0x003D	20.31	3.89	24.04	3.99	30.01	5.14
0x009D	23.18	4.03	26.23	4.22	31.43	5.31
0xC02C	20.19	2.91	23.22	3.58	27.33	4.35

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
