# Peer review of "A Role-Based Access Control Model in Modbus SCADA Systems. A Centralized Model Approach"

_sensors, 2019, doi:10.3390/s19204455_

Round 1
Reviewer 1 Report
This article outlines a security protocol to extend Modbus TCP with
role-based access control through X.509 certificates and a server-side role
database. The article is very shallow and thus difficult to follow. The lack
of technical detail, formalisation, and discussion makes impossible to
assess the security of the described protocol. Specifically, I have the
following criticism w.r.t. the submission:
- The paper does not detail its system and attacker model. Against which
attacker abilities is the protocol meant to be a defence? I vaguely
assume that the authors care about Dolev-Yao network-level attackers but
not software-level attackers? Is this correct? Is this a reasonable
attacker model?
- The evaluation only considers performance characteristics of the security
solution. A security evaluation is not done. Does the solution mitigate
the security issues outlined in the introduction and in the related work?
- The protocol is only given as a sequence diagram with no further
discussion, formalisation, and security proof. No security arguments for
the different steps of the protocol are given.
- Role-base access control for Modbus has been discussed before, e.g., in
Graham, J., Hieb, J. and Naber, J., 2016, June. Improving cybersecurity for
industrial control systems. In 2016 IEEE 25th International Symposium on
Industrial Electronics (ISIE) (pp. 618-623). IEEE.
Minor issues
-----------------------
- l.22: the abstract does not detail the "security problems of the
Modbus protocol"
- throughout: "on the other hand" is used in contrasting juxtaposition. The
phrase cannot be used without being preceded by "on the one hand".
- references should be sorted alphabetically.
Author Response
Dear reviewer,
Thank you for the review and your comments. We consider that after the first round of reviews and some changes carried out, this manuscript has improved in content and value for the reader. We hope that the new information, references, demonstration, figures, tables and analysis will meet your expectations.
We attached a document with our detailed comments.

Reviewer 2 Report
The paper proposes a role based access mechanism to increase the inherent security of the Modbus protocol. The paper is in general well-written and well structured, with a good amount of literature references. The topic of this work is certainly timely and of interest to the community of the journal.
My main concern of this current version is the following:
1) it does not become too clear how the simulation is executed. Especially, since Modbus TCP is a probabilistic access scheme, in my point of view the simulation runs should be executed multiple times per cipher suite and then the mean plus the confidence intervals should be shown when the latencies are discussed in chapter 5. As is, I am not sure these runs have been executed multiple times to account for the probabilistic nature of this protocol, in contrast to the considered deterministic benchmarking times.
1b) Please specify more carefully the setup of your simulation, for better understandability as well as for reproducability.
2) The introduction and motivation is mainly about the security shortcomings of Modbus TCP and the case study mainly presents latency results. In my point of view these two objectives should be better linked.
3) The paper unfortunately contains a large amount of small english grammar mistakes, which should be corrected before resubmitting the paper. (See below)
minor comments:
p1,l12: .. that enabel -> that enables
l13: Modbus lack -> Modbus lacks
l24: into authorization -> into an authorization
l25: an authentication process -> and an authentication process
l27: 'enables to demonstrate the proposal' -> rewrite not clear
l 35 ->and keeping -> and keeps on
l40: the Modbus protocol
l41: Figure 1 ... based on the ISA 95 model
p2 l48: can be said ... the Modbus protocol
l53: not on the field device
p3 l 65: without considering .... As is shown
l 68: the Modbus TCP/IP ... changed the Slave ID field
l 71: does not include
l 82: Please add to caption of Figure 3
l 83: the protocol -> which protocol?
p4 l95: has abbreviation DoS been introduced?
l96: provide -> integrate / include?
l102: the implementation
l107: need to adopt
l110: an authorization and ... an authentication process
l 113: The authentication proces .... -> not a sentence verb missing
l115: One important .... -> not a sentence, what exactly is the requirement?
I stopped correcting english spelling at this point. Please thoroughly proofread your paper.
p.5 l. 162: What do you mean by specification context?
l169: Not sure what the first sentence means.
l200: The authors assume the existence of an out-of-band mechanism which is however not implemented. This assumption at least requires some explanation.
p9l275: Rewrite first sentence. Meaning not clear
Author Response

(The authors gave the same response as above.)

Round 2
Reviewer 1 Report
This article presents security extensions to Modbus TCP, enabling TLS-based authentication and role-based access control through X.509 certificates and a server-side role database. The authors provide a security and performance analysis of their approach and implementation. The work appears to be technically sound and is comprehensible.
[This is an update to my previous review of this submission. The authors have addressed my concerns recommendations.]